# Using Respondent-Driven Sampling (RDS) to Identify the Healthcare Needs among Women of Reproductive Age Who Migrated from Venezuela to Brazil, 2018–2021

**DOI:** 10.3390/ijerph21060811

**Published:** 2024-06-20

**Authors:** Celia Landmann Szwarcwald, Paulo Roberto Borges de Souza Junior, Thaiza Dutra Gomes de Carvalho, Rita Suely Bacuri de Queiroz, Euclides Ayres de Castilho, Maria do Carmo Leal

**Affiliations:** 1Institute of Communication and Scientific and Technological Information in Health, Oswaldo Cruz Foundation, Av. Brasil, 4365 Manguinhos, Rio de Janeiro 21040-900, RJ, Brazil; pborges1@gmail.com; 2National School of Public Health, Oswaldo Cruz Foundation, Leopoldo Bulhões Street, 1480, 8° Floor, Bonsucesso, Rio de Janeiro 21041-210, RJ, Brazil; thaizagomes@hotmail.com (T.D.G.d.C.); ducaleal@gmail.com (M.d.C.L.); 3Leônidas and Maria Deane Institute, Oswaldo Cruz Foundation, Street Terezina, 476 Adrianópolis, Manaus 69057-070, AM, Brazil; ritabacuri@uol.com.br; 4Department of Preventive Medicine, School of Medicine, University of São Paulo, Av. Dr. Arnaldo, 455, 2° Floor, São Paulo 01246-903, SP, Brazil; castil@usp.br

**Keywords:** migrants, respondent-driven sampling, healthcare

## Abstract

In 2021, an RDS survey was conducted among Venezuelan migrant women of reproductive age who migrated to two Brazilian cities (Manaus and Boa Vista) from 2018 to 2021. To start the RDS recruitment, we chose seeds non-randomly in both cities. The study variables were age, educational level, self-rated health, pregnancy, migratory status and use of health services. We estimated the prevalence, confidence intervals and homophily effects by variable category. We used a multivariate logistic regression model to identify the main factors associated with healthcare use. A total of 761 women were recruited in Manaus and 1268 in Boa Vista. Manaus showed more irregular migrants than Boa Vista. The main reasons for using health services were as follows: illness, disease prevention and prenatal care. The logistic regression model showed the use of health services was associated with educational level and healthcare needs but not with migratory status. The social inclusion of Venezuelan migrants is extremely relevant, although many challenges must be overcome. The strategy of the Brazilian Federal Government for providing humanitarian assistance to Venezuelan migrants should be expanded to include and facilitate their integration into labor markets, access to healthcare and education, benefiting both migrants and the Brazilian people by reducing social inequality.

## 1. Introduction

Since 2014, more than 7 million Venezuelans have migrated, and of this number, more than 6 million are hosted in Latin America and the Caribbean [1]. Migration between countries in the Global South is seldom studied and poses greater challenges in receiving people from other countries due to difficulties in adequately meeting the basic social needs of local populations.

This massive migration was motivated by the unprecedented political, economic and social crisis experienced by Venezuela [2]. Multiple causes, such as the slowdown in the economy, falling oil prices, economic embargos, threats of political interventions and internal problems, which culminated in a lack of work opportunities [3], have resulted in a scenario of significant poverty, food insecurity [4], disease [5] and violence [6].

Since 2017, more than one million Venezuelans have entered Brazil, most of them across the border with Roraima state in the northern region. Of these, around 560 thousand continue to reside in Brazil [7].

In 2018, the Brazilian Federal Government organized the reception of Venezuelan migrants and refugees at the country’s northern border with the creation of the Humanitarian Response Operation “Acolhida” when they arrive in Brazil. Reception includes documentation, vaccination and the provision of shelter, food and medical assistance, as well as transferring migrants to several different municipalities with fewer Venezuelans to reduce pressure on border-state services. These actions involve the participation of 11 ministries, civil society organizations and international organizations [8].

Previous studies with Venezuelan migrants in South American countries have shown that access to healthcare services is more complicated among migrant populations. Social integration, language barriers and discrimination are factors associated with inequality in healthcare delivery [9], and these factors are also associated with worse health outcomes [10]. Furthermore, illegal migrants face even more difficulties in accessing medical care due to their immigration status and lack of documentation in the host country [11].

Regarding migrant women of reproductive age, the results of a qualitative analysis in Brazil, Ecuador and Peru showed that migrant pregnant women received hostile treatment by health professionals when they sought care during pregnancy [5]. In Colombia, migration was associated with poorer prenatal care and more common adverse perinatal events in comparison to the scenario for native Colombian women [12].

In Brazil, the Public Health System is universal and free of charge to all people [13], including migrants. Among the main reasons for the migration of Venezuelans to Brazil is access to healthcare [14,15].

In a survey carried out among Venezuelan migrant women of reproductive age in the state of Roraima, Brazil, a significant proportion of women reported being unable to obtain contraceptive methods, and almost a quarter of pregnant women did not receive prenatal care [16].

Although all pregnant women, regardless of nationality and migratory status, have the right to access public health services during pregnancy, childbirth and postpartum in Brazil [17], it is important to understand the health conditions of Venezuelan migrants of reproductive age and their needs in order to provide the appropriate healthcare.

As part of the “Redressing Gendered Health Inequalities of Displaced Women and Girls in Contexts of Protracted Crisis in Central and South America—(ReGHID)” project, coordinated by the University of Southampton, the United Kingdom, we conducted the “Survey on sexual health and reproductive health of Venezuelan migrant women in Brazil” in 2021. This project was approved by the Research Ethics Committee of the Federal University of Maranhão, logged under protocol number 35617020.9.1001.5087.

The survey originated from the need to study women of reproductive age who migrated from Venezuela to Brazil from 2018 to 2021. Collecting information from this population group is essential to monitor their health conditions and identify problems with access to and the use of health services to support improvement strategies [14,18].

Venezuelan migrant women fall into the “hidden populations” category, as there is no sampling list, and therefore the size and boundaries of this population are unknown. Additionally, there are well-founded concerns regarding secrecy, especially in the case of irregularity in the migration process [19].

Based on the assumption that Venezuelan migrant women are connected as part of a social network and have ties with other migrant women [20], we applied the Respondent-Driven Sampling (RDS) approach in order to survey this population group.

The present study aimed to identify the sociodemographic characteristics, migration patterns and health needs of Venezuelan migrant women aged 15 to 49 years old and to investigate their association with the use of health services in the 15 days prior to the survey.

## 2. Materials and Methods

An RDS survey was carried out in 2021 among Venezuelan women, aged 15 to 49 years old, who had migrated to two Brazilian cities, Manaus, the capital of the state of Amazonas, and Boa Vista, the capital of the state of Rondônia. Both states are part of the Amazon region, located in the north of Brazil on the border with Venezuela. Manaus is the largest city in the northern region, with a population of 2,300,000 inhabitants. Boa Vista is the capital of the state of Roraima and has a smaller population (440,000 inhabitants). Both cities are about 1500 km from Venezuela.

### 2.1. The RDS Method

The RDS method was proposed in 1997 by Douglas Heckathorn [21]. It is considered a variant of the chain sampling method since members of the population group under study recruit their own peers to participate. Unlike other non-probabilistic chain sampling methods, such as the so-called “snowball” [19] method, RDS is implemented under statistical assumptions that enable the calculation of selection probabilities [22,23].

When applying the RDS method, data are collected through successive recruitment cycles, called “waves”. First, individuals from the target population, called “seeds”, are selected in a non-random way to participate in the study to start the recruitment process. The seeds are asked to recruit a fixed number of individuals among their acquaintances from the same population subgroup. Individuals recruited by the seed then recruit other individuals and so on until a sufficient sample size is reached to establish equilibrium for all variables of interest [21].

A participant’s tendency to recruit individuals with similar characteristics is called the “homophily effect” [23]. Recruitment using RDS is modeled by a Markov process that takes into account the bias related to the non-random selection of individuals and the possible overrepresentation of certain characteristics in the population [24]. Furthermore, the Markov process is a stochastic process classified into finite and discrete states, whose probability of transition from one state to another depends only on the current state [24].

In the context of recruitment using the RDS method, this means that recruitment memory is given wave by wave so that the characteristics presented in the recruited individual depend only on those of their recruiter and not on the recruiter of their recruiter or on any participant of previous waves. After enough waves, the characteristics of the individuals in the final sample are independent of those of the seed. Therefore, the sampling process needs to consist of a sufficiently large number of waves for the Markov equilibrium to be reached, that is, when the prevalence of the variable of interest remains constant from a certain wave onward [22].

### 2.2. The Survey

We estimated the minimum sample size at 730 valid interviews per city, calculated to estimate the percentage of migrants who received health-related care in the 15 days prior to the survey. According to the National Health Survey carried out in Brazil in 2019, we considered a proportion of 19% [25], a bilateral error of 4%, 95% confidence and a design effect of 2, as suggested in a previous publication by Salganik and Heckathorn [22].

The eligibility criteria were (i) to be a woman, (ii) to be Venezuelan, (iii) to be aged 15–49 years old (iv) to have been in Brazil for maximum of three years, (v) to present a valid coupon to participate in the study and (vi) to not have participated in the study previously.

We selected six and nine seeds non-randomly in Manaus and Boa Vista, respectively. The migrants who were chosen were well connected in the migrant community and reported having extensive social networks. In addition, they represented different age ranges, educational levels and migratory statuses to assure diverse individual characteristics in the target population.

To start the study, each seed received three coupons to be used to invite three known migrants (peers). The recruits who were invited to participate constituted the first “wave” of the study. After participating in the study, they, in turn, received three new coupons to invite three peers. This process was repeated until the minimum sample was reached in each city.

The participants’ invitations were managed through an ACESS program that included an eligibility form and a unique code assigned to each participant. The code made it possible to identify all the connections between the recruiter and the recruits.

### 2.3. Fieldwork

The Venezuelan women’s participation began with an in-person interview, with field supervisors to verify their eligibility and validate the coupons using barcodes. Initially, questions were also asked to measure the size of the participant network: “How many Venezuelan women who moved to Brazil in the last three years do you know personally, meaning do you know each other by name? How many of them are between 15 and 49 years of age?”.

We developed the questionnaire on tablets using the RedCap—Research Electronic Data Capture system [26], which allowed for daily supervision of the collected information. The Oswaldo Cruz Foundation (Fundação Oswaldo Cruz-Fiocruz, in Portuguese) team prepared the questionnaire, which was applied in Spanish by Venezuelan interviewers who had been previously trained by the Fiocruz team. The questionnaire included the following modules: sociodemographic characteristics, migration process, work situation and income. Questions were also asked on health-related costs, health behaviors and access to and use of health services. Specific questions related to women’s health were asked regarding reproductive history, prenatal care, childbirth and abortion. The interview was conducted in a secluded place so that the participant had the privacy to answer all types of questions.

### 2.4. Study Variables

In this study, the following variables were considered: age range (15–24; 25–34; 35–49), educational level (up to elementary school, up to high school, college education), migratory status (resident, asylum seeker, irregular), pregnancy (being pregnant at the time of the survey (yes, no)), self-rated health (good, fair, poor) and the use of health services in the fifteen days prior to the date of the survey (yes, no), as well as the main reasons for using health services (illness or ongoing disease treatment, prevention, medical check-ups or childcare, vaccination, prenatal care or complementary diagnostic exams).

### 2.5. Data Analysis

The sample weighting was inversely proportional to each participant network size under the assumption that the larger the size of the participant network, the greater the chance of being invited to participate in the study [22]. However, studies have discussed other forms of data weighting [27,28]. As information provided by survey participants, the number of acquaintances of a person can be highly biased. Therefore, a range of variation from 3 to 150 was stipulated for the size of the network [29]. Seeds were excluded from the analysis, as they were selected according to directed choice [22].

To show the effects of homophily, for all the study variables, the estimates of the conditional proportions of their recruiters given each variable category were compared to the total sample proportional distribution by the variable categories. We used the chi-square statistical test for the homogeneity of the recruiters’ conditional distribution and the marginal distribution to check whether the homophily effects were significant at the 5% level.

Taking into account the effect of homophily or dependence on the observations, the design of the data collection using RDS was considered a cluster sample, with each cluster made up of the recruits of each recruiter [30]. This procedure is analogous to RDS-II in the RDS-Analyst software, suitable for populations of an unknown size [31]. The variance in the indicators of interest was estimated using a bootstrap procedure through several simulations of samples generated by the same process that originated the total sample [32].

We considered the complex recruitment RDS design [21] to estimate the prevalence and its 95% confidence interval of women who migrated from Venezuela to Brazil according to age group, level of education, self-rated health, pregnancy, migratory status and use of health services in the last fifteen days prior to the survey, as well as the main reasons for using the health services.

We used the chi-square statistical test to investigate differences between the variable distributions in the two cities (Manaus and Boa Vista).

Considering the use of a health service in the fifteen days prior to the survey as the response variable, we applied bivariate logistic regression models to test the association with all the other study variables and a multivariate logistic regression model to identify the main factors associated with the outcome.

All the analyses were performed using SPSS software version 23.0 (IBM SPSS Statistics for Windows, Version 23.0. Armonk, NY, USA: IBM Corp.) and considering the RDS complex sampling design.

## 3. Results

Migrant networks according to migratory status are shown in Figure 1. In both cities, most people are already residents in Brazil. However, there are a large number of migrants in an irregular situation in Manaus, whereas in Boa Vista, most of the Venezuelan migrants have applied for asylum in Brazil.

The results presented in Table 1 show that women who had completed high school predominated both in Manaus and Boa Vista, although migrants in Manaus had a higher level of education compared to those in Boa Vista. While there were fewer women who had only elementary-level schooling in Manaus, those with college education were less prevalent in Boa Vista.

The homophily effects for each of the variables are also shown in Table 1. There was no significant homophily effect for age in the two cities, i.e., the tendency of Venezuelan migrants to recruit others in the same age group was not significant. The conditional age distribution is almost the same as the total for the younger age groups (15–24 and 25–34). However, there was a reverse effect among older women (≥35 years), as they invited women from the young age group to a greater extent than the total, especially in Manaus.

In Manaus, migrants with a lower educational level invited a higher percentage of women at the same educational level (17.4%) than that found in the total sample (12.2%). Conversely, the percentage of women in Boa Vista with higher education who invited women with an elementary school education was 12.6%, lower than the total percentage (17.9%) (Table 1), but the homophily effects were not significant at the 5% level in both cities.

Regarding migratory status, in both cities, participants in an irregular situation invited a large proportion of migrants in the same situation. In Boa Vista, the percentage of people recruited in an irregular situation by recruiters in a similar situation was 19%, much higher than that found in the total sample (5.1%), with a significant homophily effect (*p* < 1%) (Table 1).

There was no homophily effect regarding the use of health services in the last fifteen days in either city, with similar estimates of conditional probabilities to the marginal ones (Table 1).

Table 2 shows the percentage of women estimated in each variable category in each of the two cities. The comparative analysis of the distribution by age group of Venezuelan migrants in Manaus and Boa Vista reveals very similar distributions, and no significant difference was found. As for the level of education, there is a significant difference between the proportional distributions in the two cities (*p* < 1%). In Manaus, the percentage of migrants with higher education was 21.2% (95% CI: 18.2–24.4%), twice that found in Boa Vista, at 10.3% (95% CI: 8.1–12.5%).

Significant differences in the two cities were also found for good self-reported health, with percentages of 66.4% (95% CI: 62.4–70.1%) in Manaus and 74.9% (95% CI: 69.3–77.9%) in Boa Vista. No differences were found for pregnancy at the time of the survey, with similar percentages (around 6%) in the two cities.

Different scenarios in terms of migratory patterns in the two cities were found (Table 2). The percentage of asylum seekers in Manaus was 29.4% (95% CI: 25.7–33.0%), and in Boa Vista, it was 52.1% (95% CI: 48.4–55.8%). The proportion of migrants in an irregular situation in Manaus was 26.0% (95% CI: 22.0–30.0%), significantly higher than the percentage in Boa Vista, at 5.1% (95% CI: 3.2–7.0%).

In Table 3, we present the main healthcare needs among Venezuelan women of reproductive age who have migrated to Manaus and Boa Vista. Among the main reasons for using a health service in the last 15 days are illness or ongoing treatment of a disease, prevention, medical check-ups or childcare, vaccination, prenatal care and complementary diagnostic exams (blood, urine, imaging, etc.). In Manaus, the main reasons for using a health service are disease prevention, medical check-ups amd childcare, while in Boa Vista, they are having an illness or ongoing treatment of a disease. Use of prenatal care of approximately 13% in both cities represents the needs of pregnant women at the time of the survey.

Another important health need of Venezuelan women is expressed by the high proportions of women who have a non-communicable chronic disease (18.9% in Manaus and 16.5% in Boa Vista). Additionally, more than a quarter of Venezuelan women have fair/poor self-rated health (Table 3).

Table 4 presents the results of the association of the use of health service in the last 15 days with each study variable. In both cities, this outcome was significantly associated with self-rated health, greater among migrants who reported a poor health status, and with pregnancy at the time of the survey. In Manaus, the use of healthcare was associated with level of education. In both cities, no significant association of this outcome with migratory status was found.

The multivariate logistic regression model results showed that the main factor associated with the use of healthcare in the last 15 days was pregnancy at the time of the interview after controlling for all the other study variables, both in Manaus (OR = 5.57, 95% CI: 2.57–12.04) and in Boa Vista (OR = 2.04, 95% CI: 1.13–3.68). The odds of using healthcare were about 1.5 times higher among migrant women who reported a fair or poor health status in both cities. In Manaus, migrants with secondary education were statistically less likely to use health services than those with higher education (OR = 1.72; 95% CI: 1.14–2.60) (Table 5).

## 4. Discussion

As found in previous studies involving migrant individuals [33,34,35,36], the hypothesis that Venezuelan migrants are connected in social networks and have ties with other migrants was confirmed. Networks in both cities developed quickly. Recruitment in Manaus lasted less than two months, and 761 women participated, 6 of whom were seeds. The acceptance of the survey was also very good, with an average of 2.6 recruits per recruiter. Recruitment in Boa Vista lasted less than one month, reaching a sample of 1268 participants, including 9 seeds. Although it was faster, acceptance of the survey was a little worse, with an average number of recruits per recruiter close to 2.

Moreover, the Venezuelan migrant population represents a very small percentage of the general Brazilian population. Using traditional sampling methods, very large samples would be needed to gather a sufficient number of migrants to make it possible to estimate reproductive health indicators, disaggregated by variables of interest, such as age, educational level and place of residence, which would not be feasible due to operational difficulties and costs [37,38]. During the COVID-19 pandemic, a survey implemented RDS strategies to assess the challenges faced by Venezuelan refugees, and the methodology was suitable at sites where the sample size was reached [36]. A study that collected health information from Syrians who had migrated to Germany concluded that the RDS method was more appropriate and had a much higher response rate than that achieved by random sampling of population records [34].

A complication in identifying migrant women in Brazil in sampling surveys is the assumption that migrants located in shelters and reception institutions are representative of the universe of women who have migrated from Venezuela to Brazil. This type of approach reveals clear limitations in obtaining a representative sample. The greatest challenge is including migrants who do not enter these institutions, such as refugees or those who are affiliated with another type of organization, such as religious or civil society organizations.

One limitation of RDS recruitment lies in its design. Since the development of networks is driven by the participants themselves, the characteristics of the surveyed individuals may be similar, implying a bias in estimating the proportional distributions of the variables of interest [39]. In this study, the only variable with a significant homophily effect was migratory status, as there was a large percentage of participants in an irregular situation recruited by migrants in the same situation. Regarding the other variables, the homophily effects were diluted by the expansion of the networks across several waves of recruitment, converging in the composition of a large, comprehensive sample of population characteristics [40]. Together with the stratification of the seeds by sociodemographic characteristics, these factors were important to achieve diversity in the sample.

The analysis of the percentages in each variable category revealed that Venezuelan migrants of reproductive age are concentrated in younger age groups (15–24 and 25–34) both in Manaus and Boa Vista. According to the National Health Survey conducted in Brazil in 2019 [41], the age distribution of Venezuelan migrants is much younger than that of Brazilian women aged 15 to 49 years old living in both cities. The percentages of Brazilian women aged 35–49 years in Manaus and Boa Vista are about 10 percentage points higher than that of Venezuelan migrants.

As for educational level, although migrants in both cities are most frequently educated up to high school, there is a higher proportion of women with college education in Manaus than in Boa Vista. Additionally, 21.2% of Venezuelan women living in Manaus have a college education, close to the proportion found among Brazilian citizens residing in this city (20.6%). However, in Boa Vista, female Venezuelans have a lower level of education than female Brazilians. There, 25.5% of Brazilian women of reproductive age have completed college-level education [41], but this percentage is 10.3% among Venezuelan migrants.

Time of residence in Brazil is a plausible explanatory variable for the better level of education among migrants residing in Manaus. The Venezuelan migrants who live in this city have been in Brazil longer and immigrated with a better level of education. There has been a process of “migratory impoverishment”, and the women who migrated to Boa Vista have been in Brazil for less time and have a lower level of education [14].

When comparing the Venezuelans who currently reside in Manaus and Boa Vista, the most significant difference was in the distribution by migratory status. Even though the percentage of resident women was similar in both cities, the differences occurred in situations of irregularity and asylum requests. While more than one-quarter of the women who moved to Manaus were in an irregular situation, the percentage in Boa Vista was only 5.1%. The percentage of asylum requests in Manaus was almost 30%, while more than 50% of the residents in Boa Vista had already requested asylum. This scenario reveals the considerable difficulty of regularizing the migratory situation in Manaus. In Boa Vista, most of the women live in shelters provided by the Brazilian government, possibly making it easier to obtain the necessary documentation to regularize residence in Brazil.

Self-assessment of health has been widely used to describe the health needs of population groups. In addition to health status and its relationship with clinical conditions and morbimortality indicators [42], individual perception of health involves physical, emotional well-being and life satisfaction components [43]. In this survey, more than a quarter of migrant women in both cities, although young, reported a fair or poor health status, probably a reflection of difficulties during migration, pre-existing health problems they had in Venezuela that were exacerbated during the migratory process or even challenges related to social integration, discrimination and adapting to living in a different country [9].

Although the Brazilian government closed the border during the COVID-19 pandemic, migration continued through informal routes, which is likely to have affected self-rated health [16]. Chile also experienced a significant influx of Venezuelan women via illegal crossing points, which led to a series of adverse events and important challenges in terms of health outcomes [44]. Impacts of the COVID-19 pandemic on migration and access to healthcare were also seen among Venezuelan migrants in Colombia [45].

Regarding the utilization of health services in the fifteen days prior to the survey, the results of this study are noteworthy. Firstly, the percentage of use was high in both cities, i.e., over 30%. According to data from the 2019 National Health Survey [38], this percentage was approximately twice that found among Brazilian women residing in Manaus (16.5%) and Boa Vista (14.2%), which is probably explained by the shortage of healthcare in the country of origin. Secondly, the findings also indicate that Venezuelan migrants have more healthcare needs than Brazilian women, as the lack of medical services in Venezuela has made the migrant Venezuelan population more vulnerable and susceptible to various diseases [46].

In both cities, the use of healthcare services by Venezuelan women was associated with the healthcare needs necessitated by fair/poor self-assessments of health and being pregnant at the time of the survey. In fact, in both cities, the main reasons for using health services in the last 15 days were having a health problem, disease prevention and vaccination and prenatal care. Migrant pregnant women are considered highly vulnerable and to have greater healthcare needs in terms of monitoring pregnancy. Studies have shown that migration during pregnancy may result in more adverse maternal and perinatal outcomes [5,10].

The multivariate logistic regression model results in Manaus showed inequality in the use of health services by level of education, with greater use among women with higher education, and it may be that they are more easily able to obtain information about rights and how to access health services. Nevertheless, the use of healthcare by Venezuelan migrants in Brazil was not associated with migratory status after controlling for age, level of education or healthcare needs. This is an important finding because it shows that even women in a situation of irregularity in the migration process have unrestricted access to the Brazilian Unified Health System (SUS) and free care, as do all Brazilian citizens [17].

One limitation of this study is that the time of residence in Brazil was very short for most of the migrants. Especially in Boa Vista, 83% of migrants had lived in the country for less than a year at the time of the survey. Indeed, immigrants adapt to the lifestyle in the host country over time and according to their life experience in that country [47]. Therefore, the results of this study are a reflection of a difficult migratory process, complications brought about by the COVID-19 pandemic and the consequent public health crisis [48] and little time to adjust to life in Brazil.

Among other caveats, we must cite the limitations of RDS. There is concern over whether the group of people reached in the sample using networks represents the population of Venezuelan women who have migrated to Brazil. Although the recruitment process was developed through long chains of reference, perhaps the homophily effect found among participants in an irregular migratory situation may have affected the estimates of the indicators. However, participants in an irregular migratory situation would remain “hidden” using traditional sampling methods based on migration records.

Although the results of the study should be interpreted in light of the limitations of the RDS method, its application made it possible to describe the sociodemographic characteristics, migration patterns and access to the Brazilian public health system of women of reproductive age who migrated from Venezuela to Brazil from 2018 to 2021 in search of improved living and health conditions.

## 5. Conclusions

The arrival of a large number of Venezuelan migrants in Brazil has created challenges in Brazilian society. Initially, managing the influx of Venezuelan migrants and refugees was carried out on the basis of limited migration and integration policies. During a period of intense migration, it was necessary to quickly create strategies, establishing migration policies and procedures to regularize and integrate Venezuelans. Flexible and accessible regularization processes were created in the country, mainly in Boa Vista, which provided asylum to a large number of Venezuelans [8]. However, migrants and refugees face much higher rates of unemployment and discrimination, and one of the main problems is still the social integration of Venezuelan migrants.

Nowadays, Venezuelan migrants are working informally at higher rates than Brazilians who are employed in similar activities. This high level of informality makes the migrant population more vulnerable to underemployment conditions and deprivation. Furthermore, there are difficulties in their academic and/or professional training being recognized, which restricts their access to the formal job market. The majority work in low-skilled and low-paid jobs, regardless of their level of education, limiting their employment opportunities and socioeconomic inclusion [49].

It is necessary to consider that the period of intense migration of Venezuelans coincided with a major social and economic crisis in Brazil, as well as the arrival of the COVID-19 pandemic. As has happened in many countries, the pandemic affected the income of Brazilians and overloaded the public health system [50]. Thus, the COVID-19 pandemic brought more challenges to migratory movements, exacerbating conflicts over access to the healthcare system and creating new risks to public health.

Among the contributions of the present study, we must first mention the use of the RDS method to recruit Venezuelan women of reproductive age, making it possible to describe sociodemographic characteristics, migratory patterns and access to the Brazilian public health system. Secondly, the analysis showed that Venezuelan migrant women use the Brazilian healthcare system more than Brazilian women, regardless of their migration status. These findings show the social inclusion of Venezuelan women of reproductive age in terms of healthcare use. This will contribute not only to the improvement of their general health but also to the more humanitarian development of the local health system and to the satisfaction of the health needs of the population as a whole. However, the negative effect of migration on health status highlights that it is necessary to specifically identify the needs of people with a migration history.

Ensuring access to healthcare, education and work opportunities is a fundamental human right. Social inequalities are associated with worse health outcomes, which further burdens the demand for social protection and use of health services. The strategy of the Brazilian Federal Government for providing humanitarian assistance to Venezuelan migrants should be expanded to include and facilitate their integration into labor markets, access to healthcare and education, benefiting both migrants and the Brazilian people by reducing social inequality.

Among the challenges to overcome, it is important to reduce language barriers and discrimination regarding the use of essential services [9]. Residents of the state of Roraima (located at the northern border with Venezuela) have expressed a desire to limit migrants’ access to essential services and believe that the increase in insecurity and violence is due to the intense immigration of Venezuelans into the state [51].

Understanding that many displaced Venezuelans will remain in Brazil for an extended period, if not permanently, strategies must be implemented to ensure lasting solutions to the issue of migration, such as the full integration of migrants into Brazilian society, improvement of their sense of belonging and transferring migrants from the border areas to other Brazilian states.

## Figures and Tables

**Figure 1 ijerph-21-00811-f001:**
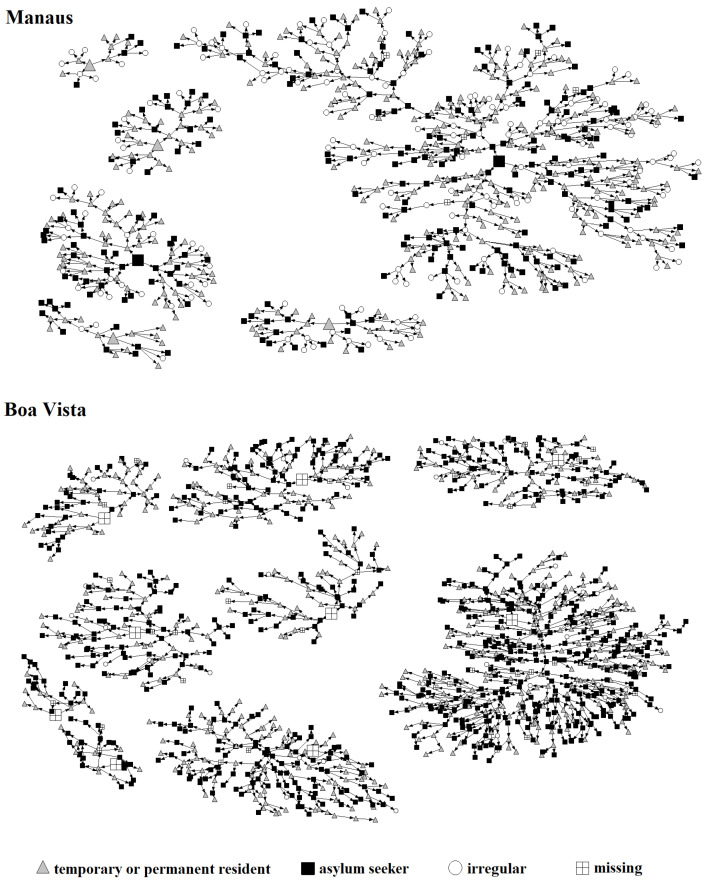
RDS networks by migratory status in Manaus and Boa Vista.

**Table 1 ijerph-21-00811-t001:** Homophily effects according to age group, educational level, migratory status and use of health services in the last 15 days among Venezuelan migrants aged 15–49 years old. Manaus and Boa Vista, 2021.

Manaus					Boa Vista				
Recruiter	Recruit				Recruiter		Recruit		
Age Group (Years)					Age Group (Years)				
	15–24	25–34	35+	Total		15–24	25–34	35+	Total
15 to24	34.6	33.0	32.3	100.0	15 to 24	38.0	35.1	26.9	100.0
25 to34	35.2	39.1	25.7	100.0	25 to 34	41.0	36.4	22.6	100.0
35+	40.2	32.0	27.9	100.0	35+	35.2	37.9	26.9	100.0
Total	36.7	35.1	28.1	100.0	Total	38.2	36.4	25.4	100.0
*p*-value *				NS	*p*-value *				NS
Educational level					Educational level				
	Elementary	High School	College	Total		Elementary	High School	College	Total
Elementary	17.4	59.1	23.5	100.0	Elementary	19.2	73.6	7.2	100.0
High school	12.6	65.7	21.6	100.0	High School	18.5	70.8	10.8	100.0
College	9.6	70.8	19.6	100.0	College	12.6	75.9	11.5	100.0
Total	12.2	66.5	21.2	100.0	Total	17.9	71.8	10.3	100.0
*p*-value *				NS	*p*-value *				NS
Migratory status					Migratory status				
	Asylum seeker	Resident ^1^	Irregular	Total		Asylum seeker	Resident ^1^	Irregular	Total
Asylum seeker	31.8	43.9	24.3	100.0	Asylum seeker	55.8	40.5	3.7	100.0
Resident ^1^	28.6	46.2	25.1	100.0	Resident	48.4	45.7	5.9	100.0
Irregular	27.5	41.7	30.8	100.0	Irregular	43.3	37.7	19.0	100.0
Total	29.4	44.7	26.0	100.0	Total	52.1	42.9	5.1	100.0
*p*-value *				NS	*p*-value *				<1%
Use of health services in the last 15 days					Use of health services in the last 15 days				
	No	Yes	Total			No	Yes	Total	
No	70.3	29.7	100.0		No	67.6	32.4	100.0	
Yes	66.5	33.5	100.0		Yes	68.7	31.3	100.0	
Total	68.9	31.1	100.0		Total	68.0	32.0	100.0	
*p*-value *			NS		*p*-value *			NS	

^1^ Temporary or permanent. * Descriptive significance level of the chi-square test for homogeneity of distributions.

**Table 2 ijerph-21-00811-t002:** Estimated prevalence (%) and 95% confidence interval by age group, educational level, migratory status and use of health services in the last 15 days among Venezuelan migrants aged 15–49 years old. Manaus and Boa Vista, 2021.

Manaus				Boa Vista			*p*-Value *
Age Group	%	95% CI		%	95% CI		
		LL	UL		LL	UL	
15–24	36.7	32.8	40.6	38.2	34.7	41.6	NS
25–34	35.1	31.0	39.2	36.4	33.1	39.7	
35–49	28.1	24.6	31.6	25.4	22.4	28.4	
Educational level	%	95% CI		%	95% CI		<1%
		LL	UL		LL	UL	
Elementary	12.2	9.5	14.9	17.9	15.3	20.5	
High school	66.5	62.8	70.2	71.8	68.6	74.9	
College	21.2	18.0	24.4	10.3	8.1	12.5	
Pregnancy ^1^	%	95% CI		%	95% CI		NS
		LL	UL		LL	UL	
Yes	5.6	4.1	7.7	6.4	4.9	8.4	
No	94.4	92.3	95.9	93.6	92.5	95.1	
Self-Rated Health	%	95% CI		%	95% CI		<1%
		LL	UL		LL	UL	
Good	66.4	62.4	70.1	74.9	69.3	77.9	
Fair	31.1	27.4	35.1	23.6	20.8	26.6	
Poor	2.5	1.6	4.0	1.5	0.8	2.6	
Migratory Status	%	95% CI		%	95% CI		<1%
		LL	UL		LL	UL	
Asylum seeker	29.4	25.7	33.0	52.1	48.4	55.8	
Resident ^2^	44.7	40.6	48.7	42.9	39.2	46.6	
Irregular	26.0	22.0	30.0	5.1	3.2	7.0	
Use of health services ^2^	%	95% CI		%	95% CI		NS
		LL	UL		LL	UL	
No	68.9	65.2	72.6	68.0	64.6	71.3	
Yes	31.1	27.5	34.7	32.0	28.7	35.3	

^1^ Pregnant at the time of the survey; ^2^ temporary or permanent. * *p*-value = descriptive significance level of the chi-square test for homogeneity of the variable distributions in the two cities (Manaus and Boa Vista).

**Table 3 ijerph-21-00811-t003:** Main healthcare needs among Venezuelan migrants aged 15–49. Manaus and Boa Vista, 2021.

Healthcare Needs	Prevalence (%)	
	Manaus	Boa Vista
Main reasons for health service use		
Illness or ongoing treatment of a disease	20.1	25.1
Prenatal care	12.9	13.2
Vaccination	19.5	15.2
Prevention, medical check-up or childcare	22.5	17.6
Complementary diagnostic exams (blood, urine, imaging, etc.)	8.0	7.6
Dental problem	2.4	4.4
At least one chronic disease	18.9	16.5
Fair/poor self-rated health	33.6	25.1
Pregnant at the time of the survey	5.6	6.4

**Table 4 ijerph-21-00811-t004:** Associations of the use of health services in the last 15 days with age group, level of education, pregnancy, self-reported health and migratory status among Venezuelan migrants aged 15–49 years. Manaus and Boa Vista, 2021.

Variables	Manaus		Boa Vista	
	Prevalence (%)	*p*-Value *	Prevalence (%)	*p*-Value *
Age group				
15–24	27.1	NS	28.4	NS
25–34	34.0		29.8	
35–49	31.1		33.7	
Educational level				
Elementary	30.4	0.014	32.9	NS
High School	27.5		29.0	
College Education	41.3		34.1	
Pregnancy ^1^				
Yes	67.4	*p* < 1%	45.0	0.020
No	28.5		29.1	
Self-Reported Health				
Good	27.1	0.021	27.6	0.019
Fair	37.4		37.8	
Poor	42.1		42.1	
Migratory Status				
Asylum Seeker	32.9	NS	30.0	NS
Resident ^2^	32.7		31.9	
Irregular	25.9		24.6	

^1^ Pregnant at the time of the survey; ^2^ temporary or permanent. * *p*-value = descriptive significance level of the test of association of the outcome with each independent variable.

**Table 5 ijerph-21-00811-t005:** Results of the multivariate logistic regression model with the use of health services in the last 15 days as the response variable and age group, level of education, pregnancy, self-reported health and migratory status as the independent variables among Venezuelan migrants aged 15–49 years old. Manaus and Boa Vista, 2021.

Manaus				
Model Variables	OR *	95% CI		*p*-Value **
		LL	UL	
Age Group				NS
15–24	0.85	0.58	1.26	
25+	1.00	-	-	
College Education				*p* < 1%
Yes	1.72	1.14	2.60	
No	1.00	-	-	
Pregnancy ^1^				*p* < 1%
Yes	5.57	2.57	12.04	
No	1.00	-	-	
Self-Rated Health				0.029
Fair/Poor	1.52	1.04	2.20	
Good	1.00	-	-	
Migratory Status				NS
Irregular	0.76	0.49	1.18	
Not irregular ^2^	1.00	-	-	
Boa Vista				
Model Variables	OR*	95% CI		*p*-value **
		LL	UL	
Age Group				NS
15–24	0.90	0.64	1.25	
25+	1.00	-	-	
College Education				NS
Yes	1.20	0.73	1.97	
No	1.00	-	-	
Pregnancy ^1^				0.018
Yes	2.04	1.13	3.68	
No	1.00	-	-	
Self-Rated Health				*p* < 1%
Fair/Poor	1.62	1.16	2.27	
Good	1.00	-	-	
Migratory status				NS
Irregular	0.80	0.33	1.93	
Not irregular ^2^	1.00	-	-	

^1^ Pregnant at the time of the survey; ^2^ asylum seeker or resident. Legend: * OR = odds ratio; 95% CI-95% confidence interval; LL—lower limit; UL—upper limit. ** *p*-value = descriptive significance level of the test of association with each independent variable after controlling for all others.

## Data Availability

The datasets analyzed during the current study are available from the corresponding author.

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
