# Peer review of "Using Respondent-Driven Sampling (RDS) to Identify the Healthcare Needs among Women of Reproductive Age Who Migrated from Venezuela to Brazil, 2018–2021"

_ijerph, 2024, doi:10.3390/ijerph21060811_

Round 1

Reviewer 1 Report

Comments and Suggestions for Authors

Comments

Immigration is a common event in the history of mankind, which has occurred since time immemorial, and although it is a controversial event or causes discomfort to local citizens. It is an event of the most important sources of cultural, genetic, and economic variation and diversity of civilization. As the authors say, “social inequalities are associated with worse health outcomes, which overload the Brazilian health system” (29-30). This principle is nothing new in developing countries, especially in primary or long-term health care and illegal Immigration. It is politically correct, but it does not touch the wounds of the political systems of the countries from which immigrants come, who are first and foremost people.

Why are there precarious living conditions in the countries of origin? What about wars or natural tragedies? What about the apparent lack of personal opportunities in the countries of origin? In addition, and it is good to say, the countries receiving people are not organized to receive immigrants, regardless of their number. The question then arises: how should countries act together (emigration and immigration) to mitigate or even solve, in order, this increasingly acute global problem? Econometric models can help, but people are not numbers or irrational people. The problem is political, although technique helps in political decision-making. This is not considered in this article.

Suggestions

This matter is a political issue and has to be tackled by organizations in the countries of exit and entry. Even the WMR (World Migration Report, 2022) published by the UN (https://worldmigrationreport.iom.int/wmr-2022-interactive/) only presents key data, net migration, COVID-19 effects, remittances (money), and highly topical migration issues. This is an aspect that should be mentioned in the article. Political Science supports this evidence and variable that is not included in the article. Without any reference to this, the article is lame.

Comments on the Quality of English Language

I do not come from a country whose official language is English.

Author Response

Dear Editor,

We thank the reviewers for the revision of the article “Using Respondent Driven Sampling (RDS) to identify health care needs among women of reproductive age who migrated from Venezuela to Brazil, 2018-2021”.

Please, find below the responses to all comments.

Reviewer 1:

We agree and appreciate the reviewer's comments. Indeed, our conclusion was very summarized, and we considered some migration issues in the revised version of the article.

In the introduction, we included possible reasons for the precarious living conditions of Venezuelan migrants (paragraphs 1 and 2, introduction section).

In the conclusions section, we discuss this point further, raising the problem of informal work and the vulnerability of migrants to exploitation of working conditions and poverty and we discuss that one of the main political problems still focuses on the social inclusion of Venezuelan migrants and their integration (paragraph 1 and 2, conclusions section).

We also describe the difficulties of migrants in obtaining jobs in accordance with their academic skills and professional experience (paragraph 2, conclusions section).

In the 3rd paragraph (conclusions section) we consider the COVID-19 pandemic and the complexity to migratory movements, exacerbating political tensions and creating new risks to public health.

In the 4th paragraph of the conclusions section, we emphasize the importance of social inclusion and integration of migrants and the need to reduce discrimination both in employment opportunities and in access to essential services.

Reviewer 2 Report

Comments and Suggestions for Authors

I enjoyed reading your article and have only a few comments for your consideration. First, I strongly suggest you edit your piece throughout to eliminate your use of passive voice. Aside from making your MS more difficult to read than it should be, it is often very difficult to discern who/what you take to be the subject of many sentences due to your use of passive constructions. Undertaking this effort would improve your article considerably in my view. Second, I wonder if you can be more precise as to the contribution and character of your findings both as those relate to relevant literature on the one hand, and to your conclusions/implications, on the other hand? That is, I think you can be both sharper and more expansive concerning how your study contributes to existing understanding of this population- whether strictly empirically in terms of health care system use and perceived health status, for example, or, in light of those concerns plus perhaps the implication of education level and health care access effects on settlement -related concerns, of those in your survey. Your conclusion is short and really says very little concerning what you take to be the links between your findings and the social inequality concerns to which you point. Can you flesh those out a bit more? Finally, your title implies you will report on the health care needs of this population but you really do not do so in any specific way? I wonder if it might be useful to revisit it accordingly to ensure it reflects what you do in this MS in analytical terms? 

Comments on the Quality of English Language

My biggest language/writing related concern with this piece is is that it is is written in passive voice. That fact makes it difficult to read while also making it even more difficult to determine often who the agent of specific actions and activities was. I suggest a thorough reworking/editing of the article to eliminate this important concern.

Author Response

Dear Editor,

We thank the reviewers for the revision of the article “Using Respondent Driven Sampling (RDS) to identify health care needs among women of reproductive age who migrated from Venezuela to Brazil, 2018-2021”.

Please, find below the responses to all comments.

Reviewer 2

Thank you for the suggestion. We reduced the use of passive voice, mainly in the methods section and in the abstract.

We included table 3 in the results section with the needs of Venezuelan migrants, as requested.

In the discussion section, we describe the precarious situation of Venezuelan migrants and the need to utilize essential services such as healthcare and education. And we emphasize the contribution of the article by showing that Venezuelan women make use of the Brazilian public system, regardless of their migratory status. Additionally, we point out the greater use of health care among migrants than Brazilian women and the concern about overloading the local health system.

We agree that our conclusion was very short. We improved the conclusions section, and we emphasized the importance of social inclusion and integration of migrants and the need to reduce discrimination both in employment opportunities and in access to essential services.

Round 2

Reviewer 2 Report

Comments and Suggestions for Authors

I think this effort works generally and offers a modest empirical contribution. I do continue to believe you should edit it to remove the remaining many instances of passive voice and for clarity. You also have one or two instances of a repeated word and several sentences I simply could not follow.

Comments on the Quality of English Language

I have shared a marked up MS with the handling editor to highlight passive voice and a number of sentences I simply could not follow. I hope that effort helps.

Author Response

Dear Editor,

We thank the reviewers for the revision of the article “Using Respondent Driven Sampling (RDS) to identify healthcare needs among women of reproductive age who migrated from Venezuela to Brazil, 2018-2021”.

Please, find below the responses to all comments.

Reviewer 2:

We appreciate the reviewer's comments and suggestions and the efforts to improve our article.

We considered all comments in the revised version of the article. The revised texts are marked in red.

Additionally, we asked a native English speaker to review our article.